

# EPI-SF: essential protein identification in protein interaction networks using sequence features

Sovan Saha[1], Piyali Chatterjee[2], Subhadip Basu[3] and Mita Nasipuri[3]

[1] Department of Computer Science & Engineering (Artificial Intelligence & Machine Learning), Techno Main Salt Lake, Kolkata, West Bengal, India

[2] Department of Computer Science & Engineering, Netaji Subhash Engineering College, Kolkata, West Bengal, India

[3] Department of Computer Science & Engineering, Jadavpur University, Kolkata, West Bengal, India

Corresponding authors
Sovan Saha, sovansaha12@gmail.com
Subhadip Basu, bsubhadip@gmail.com

## ABSTRACT

Proteins are considered indispensable for facilitating an organism's viability, reproductive capabilities, and other fundamental physiological functions. Conventional biological assays are characterized by prolonged duration, extensive labor requirements, and financial expenses in order to identify essential proteins. Therefore, it is widely accepted that employing computational methods is the most expeditious and effective approach to successfully discerning essential proteins. Despite being a popular choice in machine learning (ML) applications, the deep learning (DL) method is not suggested for this specific research work based on sequence features due to the restricted availability of high-quality training sets of positive and negative samples. However, some DL works on limited availability of data are also executed at recent times which will be our future scope of work. Conventional ML techniques are thus utilized in this work due to their superior performance compared to DL methodologies. In consideration of the aforementioned, a technique called EPI-SF is proposed here, which employs ML to identify essential proteins within the protein-protein interaction network (PPIN). The protein sequence is the primary determinant of protein structure and function. So, initially, relevant protein sequence features are extracted from the proteins within the PPIN. These features are subsequently utilized as input for various machine learning models, including XGB Boost Classifier, AdaBoost Classifier, logistic regression (LR), support vector classification (SVM), Decision Tree model (DT), Random Forest model (RF), and Naïve Bayes model (NB). The objective is to detect the essential proteins within the PPIN. The primary investigation conducted on yeast examined the performance of various ML models for yeast PPIN. Among these models, the RF model technique had the highest level of effectiveness, as indicated by its precision, recall, F1-score, and AUC values of 0.703, 0.720, 0.711, and 0.745, respectively. It is also found to be better in performance when compared to the other state-of-arts based on traditional centrality like betweenness centrality (BC), closeness centrality (CC), *etc.* and deep learning methods as well like DeepEP, as emphasized in the result section. As a result of its favorable performance, EPI-SF is later employed for the prediction of novel essential proteins inside the human PPIN. Due to the tendency of viruses to selectively target essential proteins involved in the transmission of diseases within human PPIN, investigations are conducted to assess the probable involvement of these proteins in COVID-19 and other related severe diseases.

# INTRODUCTION

Yeast and humans exhibit a remarkable degree of genetic similarity despite significant anatomical and cellular disparities. The two species shared several thousand genes being between them, even after undergoing distinct evolutionary paths for over a billion years (*Kachroo et al., 2022*). The genetic information included inside these genes is responsible for governing essential cellular activities. In the context of human biology, the malfunction or disruption of these genes can lead to the manifestation of various diseases (*Kachroo et al., 2022*). Scientists deploy system models to gain insights into biological processes due to several key factors. These factors include preserving of the building blocks of life, their simplicity of genetic manipulation, and standard protocols for replication and validation in laboratory environments (*Alberts, 2010*; *Hedges, 2002*). Despite their ease of use, yeast exhibits numerous fundamental cellular processes that are shared with humans, making it an extensively utilized model organism for fundamental scientific investigations. Numerous decades of extensive research pertaining to yeast have made a significant contribution to the comprehension of crucial conserved cellular mechanisms. This has consequently facilitated our comprehension of human biology and disorders such as cancer (*Duina, Miller & Keeney, 2014*; *Hoffman, Wood & Fantes, 2015*). Thus, this work is initially tested on Yeast PPIN followed by Human PPIN.

Humans are gradually becoming afflicted with fatal diseases like COVID-19 (*Song et al., 2020*), Ebola (*Gao et al., 2022*), and others for which a suitable course of therapy or immunization is not yet accessible. Although it takes a long time to produce a vaccination for a new disease, it is possible to experimentally test the efficacy of current medications on these disorders. However, money and labor are needed for experimental validation. So, computational methodologies are adopted to detect potential drug targets for repurposing (*Saha et al., 2024*; *Saha et al., 2022c*). This will aid the medical science community in finding current medications linked to these targets. One of the most important resources in the field of bioinformatics is the protein-protein interaction network (PPIN), which may retrieve pertinent biological data like unidentified protein functions (*Saha et al., 2014*; *Saha et al., 2019*; *Saha et al., 2018a*; *Sengupta et al., 2022*), possible protein interactions (*Kovács et al., 2019*), *etc*. PPIN is often described as an assemblage of proteins and their interconnections. The illness is hypothesized to be spread *via* these proteins and their interactions. Not all proteins are essential. Pathogens often focus on human proteins in the PPIN that have a higher degree of connections. This allows for a greater number of proteins to be transferred *via* a single protein (*Saha et al., 2021*; *Saha et al., 2018b*). These target proteins are called as essential proteins (*Saha et al., 2021*; *Saha et al., 2022b*; *Saha et al., 2018b*) in a PPIN. They function as targets for drugs and are also considered to be the highly efficient functional modules of the PPIN (*Saha et al., 2021*; *Saha et al., 2022c*; *Saha et al., 2018b*). As a result, identifying essential proteins in PPIN is crucial for identifying

prospective therapeutic targets (*Saha et al., 2022c*) linked to various illnesses. The same mechanism has been also applied in several diseases like COVID-19 (*Saha et al., 2022b*; *Saha et al., 2022c*) and others.

Although there have been substantial advances in biological research for detecting essential proteins, these techniques are not always successful, especially in complex PPIN. Therefore, computational methods (*Banik et al., 2022*; *Sengupta et al., 2019*) became popular in this area. In the work of *Banik et al. (2022)*, a rule-based refinement approach was implemented for essential protein identification. This refinement was executed by using protein complex and local interaction density information derived from the neighborhood protein properties in PPIN. In another work of *Sengupta et al. (2019)*, a novel protein prediction approach was presented, which combined several centrality metrics of PPIN to identify both hub (essential) and non-hub (non-essential) proteins. Similarly, fields like protein functions (*Fei et al., 2020*) and protein domains (*Wang et al., 2013*) were also immensely employed to identify essential proteins in PPIN. However, the majority of current prediction approaches either base their model on ML classifiers like a support vector machine (SVM) classification (*Hwang et al., 2009*), logistic regression (*Jha, Das & Saha, 2023*; *Saha et al., 2022a*), the Decision Tree model (*Jha, Das & Saha, 2023*; *Saha et al., 2022a*), the Random Forest model (*Jha, Das & Saha, 2023*; *Saha et al., 2022a*), AdaBoost classifier (*Jha, Das & Saha, 2023*; *Saha et al., 2022a*), and XGBoost (*Jha, Das & Saha, 2023*; *Saha et al., 2022a*) or centrality-based metrics like connect-between (*Hahn & Kern, 2005*), connect-close (*Hahn & Kern, 2005*), between-close (*Hahn & Kern, 2005*).

In order to identify essential proteins, *Xu et al. (2022)* introduced an efficient method called iMEPP that executed a maximization technique on key biological data such as gene expression, PPIN, and Gene Ontology (GO) activities. *Zhong et al. (2015)* combined centrality-based criteria with sub-cellular localization and fed them to an SVM-RFE model for essential protein identification. Later, the XGBGEMF model was proposed as an upgraded approach, producing a better subset of important proteins using ranking features (*Zhong et al., 2018*). However, all these methodologies lack automated feature learning techniques that were embedded in the proposed node2vec algorithm (*Grover & Leskovec, 2016*). For the determination of the proteins' degree of essentiality, a DL model was used. A novel method called DeepEP was proposed by *Zeng et al. (2019)*, which used a convolutional neural network (CNN) to extract feature data from gene expression profiles that were taken as input in form of the images. In order to forecast essential proteins, it additionally employed node2vec for PPIN-based topological information extraction. This information was combined with the earlier feature information. Using other relevant resources, such as co-expression level and co-expression pattern acquired from the RNA-seq data, a dynamically formed PPIN was built in another study by *Shang, Wang & Chen (2016)* and utilized to identify essential proteins. The accuracy of RNA-seq data was found to be higher than that of conventional microarray gene expression data (*Shang, Wang & Chen, 2016*). Employing only topological aspects of PPIN using various network centrality measures is not guaranteed to produce an appropriate forecast of essential proteins due to the recent increase in the number of noisy proteins in a PPIN. Therefore, gene expression data or subcellular localization are used as features for training and testing in other ML or

DL-based approaches. The drawback of subcellular localization is that it cannot adequately cover a large PPIN's abundance of proteins. Gene expression, however, is subject to several experimental restrictions and time series that may change throughout time.

The proposed method EPI-SF is constructed based on topological information and gene expression pattern. Since protein sequence is one of the most pertinent areas to examine for the discovery of essential proteins (*Wu et al., 2021*), the proposed approach first retrieves all of the protein sequences of yeast PPIN, which are then utilized to compute the following attributes: (1) Pseudo amino acid composition (PAAC) (order 1, traditional), (2) physico-chemical properties (PCP) and (3) amino acid composition (AAC). These features are supplied as training and test set inputs for a variety of machine learning models, including the XGB Boost Classifier (*Chen & Guestrin, 2016*), AdaBoost Classifier (*Freund & Schapire, 1996*), logistic regression (LR) (*Bacaër, 2011*), SVM classification (*Cortes & Vapnik, 1995*), the Decision Tree (DT) model (*Mitchell, 1997*), Random Forest (RF) model (*Breiman, 2001*), and Naïve Bayes (NB) model (*Hand & Yu, 2001*) to predict essential proteins. In this case, the Scikit-Python library is utilized. The technique is afterwards applied to identify novel essential proteins in human PPIN, which are later verified against the existing literature to determine their potential as therapeutic targets for COVID-19 and related human disorders. In Fig. 1, the entire process is highlighted. The major contribution of this work is the inclusion of substantial physiologically pertinent protein sequence features during the training and testing of ML models. Based on the results of the initial experiment on yeast, a prediction on the human PPIN is implemented. Further investigation has revealed that the anticipated proteins are also involved in various diseases.

## METHODOLOGY

### Data collection

UniProt (*The UniProt Consortium, 2017*) and BioGrid (*Stark et al., 2006*) databases are used in this study. UniProt is a significant central repository of proteins, protein interactions, protein functions, subcellular localization, protein domain, etc. BioGrid consists of physical and genetic interactions of many organisms like *Saccharomyces cerevisiae*, *Drosophila melanogaster*, *etc.* All machine learning models are initially deployed on yeast PPIN (extracted from BioGrid) to predict essential proteins. Yeast PPIN consists of 5,616 proteins and 52,833 interactions. Additionally, the Munich Information Center for Protein Sequences (MIPS) (*Mewes et al., 2006*), Saccharomyces Genome Database (SGD) (*Cherry et al., 1998*), the Database of Essential Genes (DEG) (*Zhang, Ou & Zhang, 2004*), and the Synthetic Gene Database (SGDB) (*Mallick et al., 2016*) are used to retrieve 1,199 essential and 4,026 non-essential proteins/genes from the yeast PPIN to create the positive and negative data samples that will be used to train and test the ML models. All the datasets are downloaded as on 1st May 2023. The estimation of prediction performance on the yeast PPIN is conducted, followed by the execution of the most effective model on the Human PPIN acquired from UniProt. According to the UniProt database, the analysis is based solely on a subset of human proteins, specifically those that have undergone a rigorous review process. This subset consists of 204,961 proteins. In both cases, protein sequences obtained from UniProt are utilized for the extraction of ML features (Fig. 1).
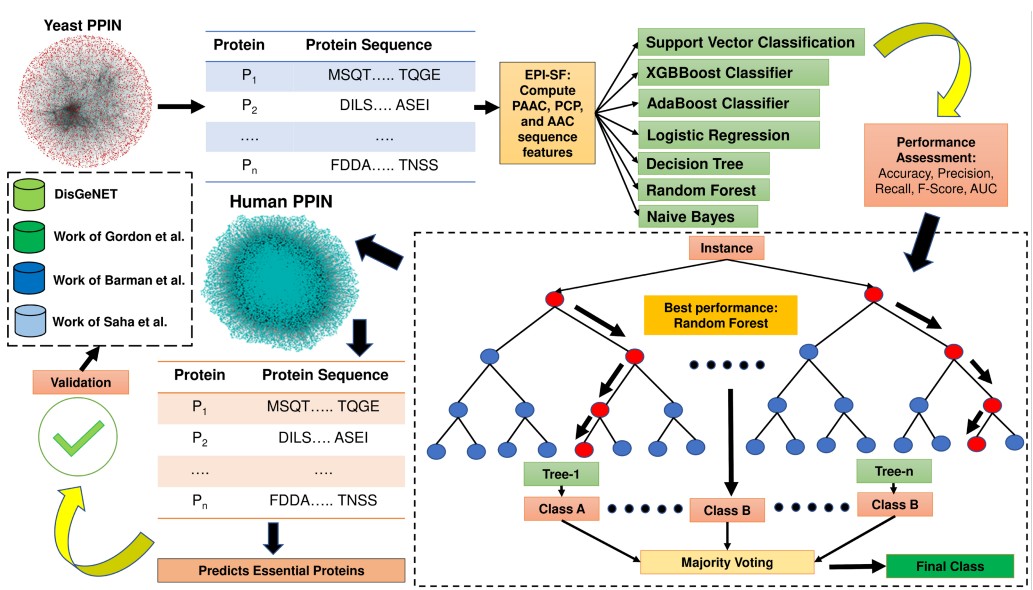

**Figure 1 Methodology of EPI-SF.** EPI-SF is initially tested on the yeast PPIN (marked in red). The best-performing model, *i.e.,* the Random Forest (RF) model, is then applied to Human PPIN (marked in green) to identify essential human proteins. The identified essential human proteins are further validated by the DisGeNET database and the work *Gordon et al. (2020)*, *Barman et al. (2022)*, and *Saha et al. (2021)*.

## Formation of positive and negative samples

A total of 1,199 and 4,026 essential and non-essential yeast genes/proteins are considered from the published papers (*Cherry et al., 1998*; *Mallick et al., 2016*; *Mewes et al., 2006*; *Zhang, Ou & Zhang, 2004*) for positive and negative classifications. These proteins are mapped with yeast PPIN proteins obtained through UniProt. The positive sample set consists of 1,199 essential proteins. To create a balanced dataset, a group of 1,199 non-essential proteins is randomly selected from the larger pool of 4,026 non-essential proteins, which forms the negative sample set. Consequently, the combined positive and negative protein samples amount to 2,398, as depicted in Fig. 2.

## Sequence feature extraction

Protein sequences from both positive and negative samples are downloaded from UniProt. To calculate feature values, they are passed into Pfeature (*Pande et al., 2022*). The protein sequences of the yeast PPIN are used to analyze three types of sequence features. They are (1) pseudo amino acid composition (PAAC) (order 1, traditional) (*Pande et al., 2022*), (2) physico-chemical properties (PCP) (*Pande et al., 2022*), and (3) amino acid composition (AAC) (*Pande et al., 2022*). In AAC, there are twenty descriptors (see Table S1), while in PCP and PAAC, there are thirty (see Table S2) and twenty-one descriptors (see Table S3), respectively (available online). The feature values of 2,398 proteins are also available online.

## Data preprocessing

Prior to training and testing machine learning models, it is essential to conduct a comprehensive study and evaluation of the acquired protein sequence feature values.
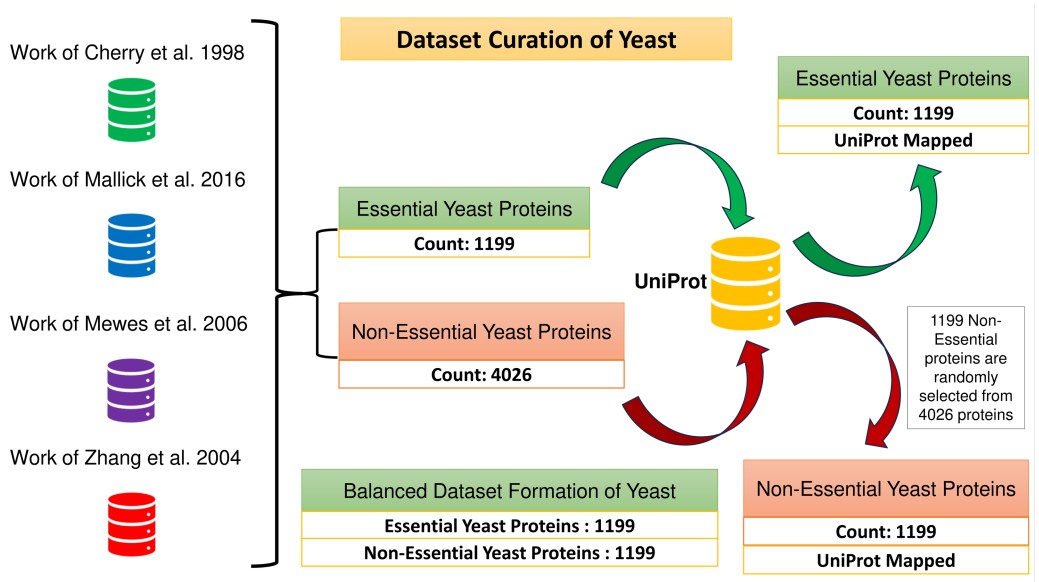

**Figure 2** Formation of the balanced dataset of yeast. The dataset contains a total of 2,398 proteins.

This is because the protein sequences obtained from UniProt are directly used as input for the Pfeature server. As Pfeature calculates all features dynamically, it is possible that some values may be omitted or overlooked owing to unavailability or delays in data fetching during calculation. Therefore, all levels of data preparation are deemed crucial in this study. It typically consists of four distinct processes, as seen in Fig. 3. To improve the comprehension of data anomalies, an analysis of the whole data structure is performed in the first stage, using statistical indicators such as mean, median, and mode. The presence of missing data in the second step can result in models selecting an inaccurate pattern and subsequently generating erroneous predictions instead of accurately identifying genuine cases. The deletion of records with missing data will lead to data loss as well. The third stage of the analysis is the identification of outliers or data points that exhibit significant deviation from the overall dataset. During the concluding phase, any identified data inconsistencies are addressed and rectified. The entire dataset is partitioned into an 80% training set and a 20% test set for the purpose of implementing different machine learning models. This is done after preprocessing the feature values of the yeast protein sequence. Yeast proteins are categorized into two distinct classification labels: (1) zero and (2) one. The value of one is used to indicate the presence of an essential protein in yeast, whereas the value of zero is assigned to proteins that are non-essential.

## Classification models

The utilization of the DL method in ML applications is widespread. However, for this particular research focused on sequence characteristics, it is not recommended due to the limited accessibility of high-quality training sets containing positive and negative samples. But since some DL works on limited availability of data are also executed at recent times, that will be our future scope of work. Several ML models have been generated and evaluated
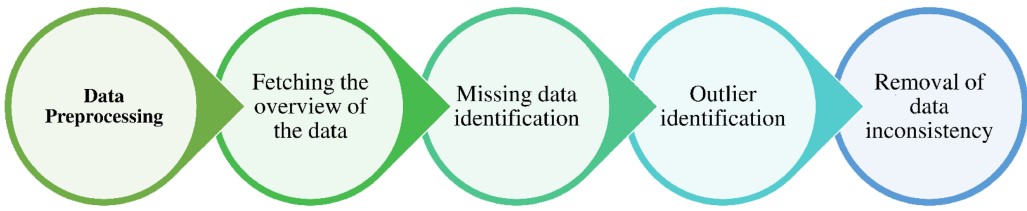

**Figure 3** **Data preprocessing phase of EPI-SF.** Four steps, each with a specific function for preparing the curated data, for use by the ML models.

using the training and test datasets derived from the features of yeast protein sequences. In this work, XGB Boost Classifier (*Chen & Guestrin, 2016*), AdaBoost Classifier (*Freund & Schapire, 1996*), LR (*Bacaër, 2011*), SVM (*Cortes & Vapnik, 1995*), DT model (*Mitchell, 1997*), RF model (*Breiman, 2001*), and NB model (*Hand & Yu, 2001*) are utilized for the purpose.

The data categorization process in supervised machine learning involves the utilization of a DT model, which employs a predetermined set of questions to classify the data. Initially, a specific attribute is selected, followed by the formulation of a query. The structure bears a resemblance to that of a tree, wherein the root node functions as the foundational element of the tree. The subsequent sequence consists of decision nodes that are interconnected by edges, with each edge indicating a distinct solution to the initial query. The terminal nodes in the decision tree symbolize the ultimate decision or the predicted class labels. RF model employs an ensemble learning technique that aims to enhance prediction accuracy by utilizing an averaging strategy. This is achieved by training multiple DT classifiers on different subsets of the dataset. DT has strong performance when used in conjunction with AdaBoost, which is a prominent boosting ensemble model. In the AdaBoost algorithm, the analysis of past errors is an ongoing process whereby the weights of data points that are deemed to have been misclassified are incrementally increased. Another strategy based on ensemble learning and Decision Trees is known as XGBoost. The framework employed in this study utilizes a gradient-boosting approach.

In comparison to preceding algorithms, XGBoost presents the advantages of regularization, parallel processing capabilities, and enhanced computational efficiency. Furthermore, it has the capability to handle missing data and incorporates an internal mechanism for prioritizing the significance of features. XGBoost has exhibited superior performance compared to other algorithms across multiple datasets, making it a commonly utilized tool in practical applications such as healthcare.

The LR model is a statistical technique that establishes a linear association between a dependent variable and one or more independent variables. The underlying assumption of the model is that the associations between the variables exhibit linearity, indicating that alterations in the dependent variable are directly proportional to corresponding alterations in the independent variable. The model aims to determine the line that optimally represents the relationship between the variables by minimizing the sum of the squared discrepancies between the observed and predicted values. SVM classify data points by utilizing a decision

boundary, also known as a hyperplane. This is an additional supervised classification model. The primary objective of the SVM algorithm is to optimize the separation between the nearest data points of each class and the decision boundary. The NB model is a classifier that can be applied to both multi-class and binary classification tasks, and it is based only on the principles of probability as outlined in Bayes theorem. It operates under the assumption of feature independence, which simplifies the calculations. It is known for its effectiveness in various practical applications, such as sentiment analysis and spam filtering, despite its inherent simplicity.

## Mapping of essential proteins of human with COVID-19 and DisGeNET datasets

The computational methodology employed for identifying protein features in humans is consistent with what has been previously mentioned in the case of yeast. According to EPI-SF using the RF model, it is projected that 5,662 out of the total 204,961 human proteins that have been examined are deemed essential (https://github.com/SovanSaha/EPI-SF-Essential-protein-identification-in-protein-interaction-networks-using-sequence-features/blob/2bbb2194072dc8cf1f24bc0128d2e7528daa7a9d/RF%20Model%20Predicted%20Ess%20Human%20Proteins.xlsx). Since pathogens target only essential proteins in PPIN to be potential baits (*Saha et al., 2018b*) hence, it is feasible to consider these identified essential proteins as crucial targets for infections to facilitate the dissemination of diseases. The 5,662 human proteins have been assigned to their respective 4,037 human genes (available online) by UniProt-ID mapper. These genes are subsequently compared with the COVID-19 human target genes identified in the studies conducted by *Barman et al. (2022)*, *Saha et al. (2021)*, *Saha et al. (2022b)* and *Gordon et al. (2020)*. A notable intersection of 1,191 genes, accounting for 30% of the total genes, has been observed (Fig. 4). This finding underscores the involvement of these genes in susceptibility to COVID-19 infection and subsequent transmission within the human body. Upon submission to the DisGeNET database (*Piñero et al., 2017*), the remaining 2,846 genes, accounting for 70% of all genes, yield a comprehensive report detailing their potential association with various human disorders.

DisGeNET serves as the primary archive for genes and variations associated with human diseases that are publically accessible. The available evidence from the DisGeNET database, specifically the gene-disease association (https://github.com/SovanSaha/EPI-SF-Essential-protein-identification-in-protein-interaction-networks-using-sequence-features/blob/037ddeb7f5dea4f0242976202ba8517b97c7d10d/Evidences%20for%20Novel%20Gene-Disease%20Association.xlsx) and variant disease association data (https://github.com/SovanSaha/EPI-SF-Essential-protein-identification-in-protein-interaction-networks-using-sequence-features/blob/a942c826e4cf2e4f1da8e3a545d2e11e9695b0de/Evidences%20for%20Novel%20Variant%20Disease%20Association.xlsx), suggests that the 2,846 genes under investigation are of significant importance. The details of these two online files are available online. Further exploration of these genes could potentially lead to the identification of human therapeutic targets for various diseases. Hence, the aforementioned set of 2,846 genes have potential as candidates for emerging as novel essential genes.

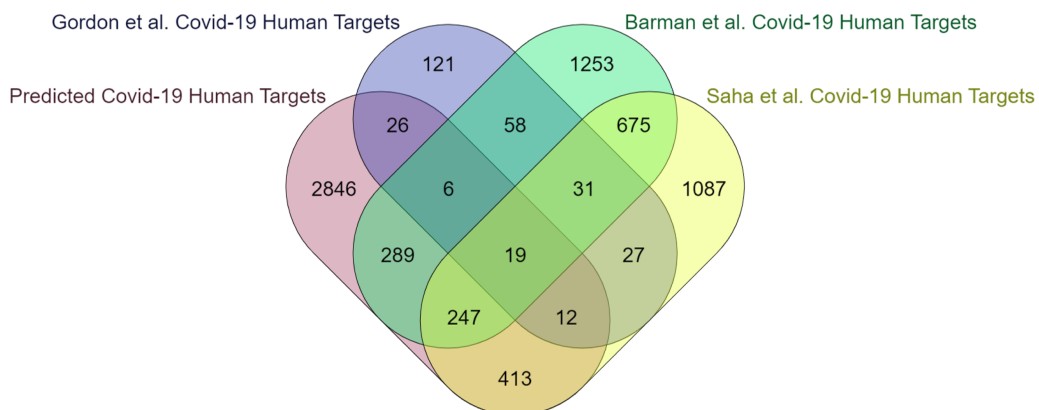

**Figure 4** The present study investigates the extent to which critical human genes revealed in previous research overlap with the human targets associated with COVID-19.

**Table 1** Performance indicator of ML models on the yeast dataset.

| ML models | Precision | Recall | F-Score | AUC |
|---|---|---|---|---|
| XGBBoost (*Chen & Guestrin, 2016*) | 0.653 | 0.740 | 0.694 | 0.735 |
| AdaBoost (*Freund & Schapire, 1996*) | 0.629 | 0.660 | 0.644 | 0.674 |
| Logistic Regression (*Bacaër, 2011*) | 0.656 | 0.712 | 0.683 | 0.698 |
| SVM (*Cortes & Vapnik, 1995*) | 0.646 | 0.740 | 0.690 | 0.702 |
| Decision Tree (*Mitchell, 1997*) | 0.593 | 0.615 | 0.588 | 0.595 |
| EPI-SF using Random Forest | 0.703 | 0.720 | 0.711 | 0.745 |
| Naïve Bayes (*Hand & Yu, 2001*) | 0.602 | 0.832 | 0.699 | 0.692 |

## RESULTS & DISCUSSION

In the initial stage of the experiment, the suggested methodology is implemented on the yeast dataset following the necessary data preprocessing procedures. The validation test set is created by partitioning 20% of the data samples, while the remaining 80% are designated as training sets. The proportion of training-validation set was consistently applied across all machine learning algorithms, including Decision Tree, Random Forest, Naïve Bayes, and others. The superior performance of EPI-SF using RF model is apparent based on the data shown in Table 1 and Fig. 5. According to the findings on the yeast dataset (Table 2), it has been observed that a significant number of machine learning models exhibit superior performance compared to other conventional essential protein prediction methods such as network-based and deep learning models. Upon evaluating the performance of several machine learning models, as presented in Table 1, it turned out that the EPI-SF using the RF model successfully predicts 5,662 essential ones out of the total 204,961 reviewed human proteins in UniProt. On further observation, it has been also noted that these 5,662 proteins are related to the cause of various human diseases like COVID-19 and others.
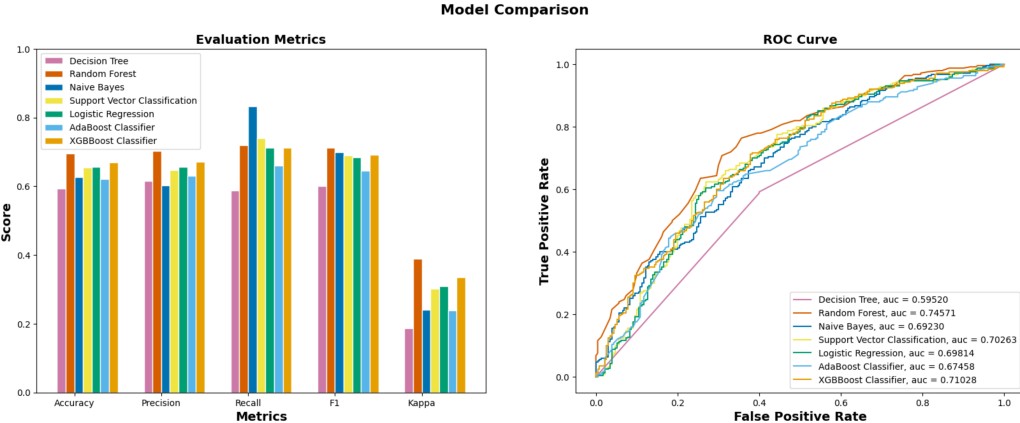

**Figure 5** **Performance metrics of ML models on the yeast dataset.** Out of seven ML models, EPI-SF using RF performs the best compared to the others.

**Table 2** **Performance indicator of centrality-based essential protein identification methodologies on the yeast dataset.**

| Methodologies | Precision | Recall | F-Score |
|---|---|---|---|
| Local Average Connectivity (LAC) (*Li et al., 2011*) | 0.471 | 0.464 | 0.599 |
| Degree Centrality (DC) (*Jeong et al., 2001*) | 0.376 | 0.363 | 0.503 |
| Density of Maximum Neighborhood Component (DMNC) (*Lin et al., 2008*) | 0.376 | 0.363 | 0.503 |
| Betweenness Centrality (BC) (*Anthonisse, 1971*) | 0.398 | 0.393 | 0.532 |
| Closeness Centrality (CC) (*Sabidussi, 1966*) | 0.266 | 0.260 | 0.391 |
| Bottle Neck (BN) (*Pržulj, Wigle & Jurisica, 2017*) | 0.381 | 0.373 | 0.512 |
| Information Centrality (IC) (*Stephenson & Zelen, 1989*) | 0.438 | 0.432 | 0.569 |
| Eigenvector Centrality (EC) (*Bonacich, 1987*) | 0.408 | 0.401 | 0.540 |
| Subgraph Centrality (SC) (*Estrada & Rodriguez-Velazquez, 2005*) | 0.408 | 0.401 | 0.540 |
| DeepEP (*Zeng et al., 2019*) | 0.580 | 0.520 | 0.550 |
| EPI-SF using Random Forest | 0.703 | 0.720 | 0.711 |

## CONCLUSION

The proposed method EPI-SF has a high level of efficiency in predicting essential proteins. The process involves the extraction of distinctive features from protein sequences, which are subsequently utilized as input for ML models in order to discern essential proteins. ML models have demonstrated superior effectiveness in the identification of essential proteins compared to classic centrality-based approaches such as LAC (*Li et al., 2011*), BC (*Anthonisse, 1971*), CC (*Sabidussi, 1966*), and others. The reason behind this arises from the reliance on centrality-based techniques, which exclusively take into account the direct or indirect interconnections around a given protein. Consequently, these techniques may not always yield valuable insights in some scenarios and may be inapplicable if the protein under investigation lacks any links. However, the problem associated with

centrality approaches can be addressed by considering the protein sequence as a primary feature, which is the main focus of this suggested study. Moreover, the protein sequence holds greater physiological significance compared to the connections in PPIN. A total of seventy-one notable features derived from three primary classifications, namely PAAC, PCP, and AAC. These classifications have been employed to construct the feature dataset using protein sequences. The yeast dataset is utilized to assess the effectiveness of the model prior to its application on the human PPIN. The predictions made by the model about human essential proteins and genes offer compelling evidence supporting their association with potential therapeutic targets for many diseases, including COVID-19. The current utility of the model is limited to the yeast and human PPIN interactome. However, there is potential for its application to be extended to additional organisms through our future research endeavors.

### Funding
The authors received support (infrastructure facilities) from the "Center for Microprocessor Applications for Training Education and Research" research laboratory of the Computer Science and Engineering Department, Jadavpur University, India. In addition, this project is also supported by the Department of Biotechnology project (No. BT/PR16356/BID/7/596/2016), Ministry of Science and Technology, Government of India. There was no additional external funding received for this study. The funders had no role in study design, data collection and analysis, decision to publish, or preparation of the manuscript.

### Grant Disclosures
The following grant information was disclosed by the authors:
Computer Science and Engineering Department, Jadavpur University, India.
Department of Biotechnology project: BT/PR16356/BID/7/596/2016.
Ministry of Science and Technology, Government of India.

### Competing Interests
The authors declare there are no competing interests.

### Author Contributions
- Sovan Saha conceived and designed the experiments, performed the experiments, analyzed the data, prepared figures and/or tables, authored or reviewed drafts of the article, and approved the final draft.
- Piyali Chatterjee conceived and designed the experiments, analyzed the data, prepared figures and/or tables, authored or reviewed drafts of the article, and approved the final draft.
- Subhadip Basu analyzed the data, prepared figures and/or tables, authored or reviewed drafts of the article, and approved the final draft.

- Mita Nasipuri analyzed the data, prepared figures and/or tables, authored or reviewed drafts of the article, and approved the final draft.

## Data Availability

The data is available at GitHub and Zenodo:

-https://github.com/SovanSaha/EPI-SF-Essential-protein-identification-in-protein-interaction-networks-using-sequence-features.git.

- Sovan Saha. (2024). SovanSaha/EPI-SF-Essential-protein-identification-in-protein-interaction-networks-using-sequence-features: EPI-SF (1.0). Zenodo. https://doi.org/10.5281/zenodo.10662829.

## Supplemental Information

Supplemental information for this article can be found online at http://dx.doi.org/10.7717/peerj.17010#supplemental-information.

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
