# Peer review of "EPI-SF: essential protein identification in protein interaction networks using sequence features"

_PeerJ, doi:10.7717/peerj.17010_

## Round 0.1 · original submission · Major Revisions

Please address the concerns of both reviewers and amend the manuscript accordingly.

**Language Note:** The review process has identified that the English language must be improved. PeerJ can provide language editing services - please contact us at copyediting@peerj.com for pricing (be sure to provide your manuscript number and title). Alternatively, you should make your own arrangements to improve the language quality and provide details in your response letter. – PeerJ Staff

Reviewer 1 ·

Basic reporting

This paper proposed an ML technique, EPI-SF, to identify essential proteins in the protein-protein interaction network (PPIN), an important aim in network analysis. Several issues can be done to improve this study.

Experimental design

It is better to add a section on how the essential proteins were mapped to COVID-19 and DisGeNET datasets.

Validity of the findings

No comment

Additional comments

There are several grammatical errors; for example, the methodology section should be written in past tense.

Several statements require reference(s), e.g., Line 87-88, 127-128, etc.

The authors may provide the importance of PPIN and essential proteins in PPIN in several diseases and give examples in the second paragraph of introduction.

Line 98: Remove “.” after modules.

Line 105: It is better to elaborate more on “other relevant works”.

Line 108-111: This sentence sounds hanging, and it is better to elaborate a little bit about ML and centrality-based method.

Line 114: Recheck the in-text citation.

Figure 1 can be placed in the methodology section.

Line 158: add the word “databases” after BioGrid (Stark et al. 2006).

It is better to add the date of the data that have been retrieved.

The full names of databases, such as SGD, MIPS, DEG, SGDB, etc.

Be careful with the abbreviation. Some abbreviations have been introduced more than one time (e.g. DT). There are also abbreviations that were written in full names.

The methods of identifying essential proteins with the associated diseases and COVID19 can be added in the methodology section.

Line 270-272: It is better to elaborate more on this sentence.

It is very useful to have an online table that contains ID, protein name, and function/ associated diseases in the results section.

What does the number in the name of the sheet of the online tables (Evidences for Novel Gene-Disease Association and Evidences for Novel Variant Disease Association) mean? The authors may provide an explanation of this.

Reviewer 2 ·

Basic reporting

No comment

Experimental design

The paper by Saha et al. discusses an important topic. While a plethora of attributes contribute to protein interactions, It is intriguing to observe that sequence-based features has been used and are shown to be sufficient to determine protein essentiality from PPINs.

1. In data preprocessing, missing data is mentioned. How can there be missing data when the features considered are all aggregate measures considered from available sequences? Overall, the data preprocessing step looks very generic and is not well explained.
2. Why was k-fold cross validation not used? The results could depend on the initial choice of the 80%-20% partitioning that has been considered.
3. The proposed method can be compared to a few other more recent methods (A quick search shows a few new methods that have cited DeepEP (Zeng et al. 2019))

Validity of the findings

No comment

Additional comments

Minor Points-

1. Write numbers with the same number of significant digits.
2. I think that- “Despite being a popular choice in Machine Learning (ML) applications, the Deep Learning (DL) method is not suggested for this specific research work based on sequence features due to the restricted availability of high-quality training sets of positive and negative samples.” is too generic and might be misleading. Generative DL doesn’t need a lot of positive and negative samples. Deep learning with limited data is also an area of active research.
3. Line 103 says- “In the work of Saha et al.” but quotes Banik et al.
4. In Table 2 providing a reference to RF for EPI-SF (the proposed method) is misleading.

---

## Round 0.2 · accepted · Accept

The concerns of the reviewers were adequately addressed, and the revised manuscript is acceptable now.

Reviewer 2 ·

Basic reporting

The authors are certainly encouraged to opt for k-fold cross validation to establish the predictive performance of their method in the future. Overall, the changes look good, and the manuscript can be accepted for publication.

Experimental design

Looks better.

Validity of the findings

Looks better.